# Input–Output-Improved Reservoir Computing Based on Duffing Resonator Processing Dynamic Temperature Compensation for MEMS Resonant Accelerometer

**DOI:** 10.3390/mi14010161

**Published:** 2023-01-08

**Authors:** Xiaowei Guo, Wuhao Yang, Tianyi Zheng, Jie Sun, Xingyin Xiong, Zheng Wang, Xudong Zou

**Affiliations:** 1The State Key Laboratory of Transducer Technology, Aerospace Information Research Institute, Chinese Academy of Sciences, Beijing 100190, China; 2School of Electronic, Electrical and Communication Engineering, University of Chinese Academy of Sciences, Beijing 100049, China

**Keywords:** reservoir computing, nonlinear MEMS resonator, algorithm optimization, dynamic temperature compensation, MEMS resonant accelerometer

## Abstract

An MEMS resonant accelerometer is a temperature-sensitive device because temperature change affects the intrinsic resonant frequency of the inner silicon beam. Most classic temperature compensation methods, such as algorithm modeling and structure design, have large errors under rapid temperature changing due to the hysteresis of the temperature response of the accelerometer. To address this issue, we propose a novel reservoir computing (RC) structure based on a nonlinear silicon resonator, which is specifically improved for predicting dynamic information that is referred to as the input–output-improved reservoir computing (IOI-RC) algorithm. It combines the polynomial fitting with the RC on the input data mapping ensuring that the system always resides in the rich nonlinear state. Meanwhile, the output layer is also optimized by vector concatenation operation for higher memory capacity. Therefore, the new system has better performance in dynamic temperature compensation. In addition, the method is real-time, with easy hardware implementation that can be integrated with MEMS sensors. The experiment’s result showed a 93% improvement in IOI-RC compared to raw data in a temperature range of −20–60 °C. The study confirmed the feasibility of RC in realizing dynamic temperature compensation precisely, which provides a potential real-time online temperature compensation method and a sensor system with edge computing.

## 1. Introduction

The MEMS resonant accelerometer has emerged as a powerful inertial sensors owing to its advantage of small size, low cost, and low power consumption [1,2,3]. However, temperature affects the performance of silicon-based sensors, so temperature compensation is crucial for providing reliable output. It is now commonly established that temperature compensation can be realized by device structure optimization at the hardware level [4,5,6], and frequency-temperature (f-T) modelling at the software level [7,8,9]. The hardware compensation methods usually require special structural topology and sophisticated fabrication technology [10], which suffer from high cost and power consumption. Thus, temperature compensation at the software level possesses broad application prospects. However, software compensation generally needs a complicated network with a mass of parameters for higher accuracy [11]. Moreover, rapid changing of ambient temperature causes hysteresis of temperature response of the accelerometer, but this dynamic case has received little attention in the literature so far.

To this end, surveys have shown that adding derivative terms in polynomial or using echo state network (ESN) can improve the precision to some extent [12,13]. Inspired by this view, combined with the characteristics of reservoir computing (RC) that is simple in structure and well-suited for temporal processing [14], we proposed a dynamic temperature compensation for an MEMS resonant accelerometer based on an improved RC structure with a single duffing resonator to realize a high-accuracy compensation. Evolved from recurrent neural network (RNN), RC is a brain-inspired computational algorithm that is best-in-class for processing information generated by dynamic systems. Even when systems display complex spatiotemporal behaviors, which are considered the hardest problems, an optimized RC can predict them easily. With the proposal of a time-delayed structure, which reduces the difficulty of hardware implementation [15], physical RC in diverse fields has been studied [16,17,18], among which the RC based on MEMS resonator has a simpler system and lower cost [19]. Many optimization methods of diverse RC structures based on MEMS resonators have also been discussed in recent studies [20,21,22]. Benefiting from these advantages, numerous tasks with various datasets, such as chaotic system [23], handwritten digit [22], and IMU signal [20], can be processed by RC, which inspires us to handle the signal processing of sensors via RC. This method is suitable for addressing the hysteresis phenomenon. Moreover, the physical RC with a single resonator makes this temperature compensation method easier to integrate with an MEMS accelerometer and IC.

This paper sets out to investigate a novel real-time compensation method and an application of physical RC system based on MEMS resonator. We apply RC to the temperature compensation for MEMS accelerometer for the first time, which not only realizes a dynamic temperature compensation, but also lays a foundation for the disruptive applications of physical RC in the field of sensor temperature compensation technology and provides a new insight for sensory–computation integration. Our method is universal and hardware implementation is easy, as the RC can be trained for different devices and environments, and our physical RC based on a single MEMS resonator is simple to operate with an MEMS accelerometer. It is hoped that this research will contribute to a new generation of temperature compensation for all kinds of MEMS sensors.

## 2. Materials and Methods

### 2.1. Input–Output-Improved Reservoir Computing

A traditional RC (T-RC) is shown in Figure 1a. In the input layer, the original data u(t_i_) are normalized to (0,1), because the resonator is amplitude-detected and the output data only reflects positive information. The original data are broadcast to the reservoir by the N-dimensional **mask** in order to increase linear richness. It acts as the input connection weight of ESN and is randomly chosen to avoid the vanishing gradient problem during training [24]. The **mask** in our algorithm is a vector of random numbers in the range of (−1, 1) with zero mean and unit variance. Then, linear feature vector **x_i_** is fed to the reservoir layer with an input gain β. The reservoir layer is the most vital part of RC, which increases the nonlinear richness of the system and is sensitive to its optimizable architecture properties. The nonlinear node, as shown in the center yellow NL square in Figure 1a, maps **x_i_** into a nonlinear feature vector **r_i_**. Each element of the vector represents a virtual node, which contains high-dimensional features deriving from the nonlinearity of the duffing function [25], described as:(1)$$m\ddot{x}{+c\dot{x}+k}_{1}{x+k}_{3}x^{3}=\mathrm{Fcos}\left(\omega t\right).$$
where m is the lumped effective mass, x is the displacement of silicon beam, c is the damping coefficient, k_1_ is the linear spring stiffness, k_3_ is the nonlinear spring stiffness that determines the nonlinear behavior, F is the electrostatic drive force, and ω is the drive frequency. The time interval θ between each element in feature vectors is the duration of each virtual nodes. It decides the duration time Nθ of a single input data u(t_i_) as well as the data sampling time before modulation in hardware experiment [22]. Through the delay loop, as illustrated by the yellow arrow in Figure 1a, previous data are added to the present data element by element and then sent to the nonlinear node, so the dynamic system is able to remember its previous states. We chose the delay time of the delay loop τ = Nθ. The feedback gain α affects the memory capacity (MC), which is crucial to accuracy. By adding delay loops (the gray arrows), we can enhance the MC of the RC system. Optimization method of those parameters can be referred to in our previous work [21]. The output layer yields the RC output by linearly weighting the reservoir feature vector: $\hat{y}\left(t_{i}\right){=\mathrm{wr}}_{i}$, where **w** is the output weight vector. For the RC system, only the **w** is trained via ridge regression algorithm:(2)$${w=yX}^{T}{{(XX}^{T}+\lambda I)}^{-1}.$$
where **X** is the feature matrix by stacking up **r_i_**, **y** is the target value, λ is regularization parameter to prevent overfitting and **I** is the identity matrix.

The main characteristic of RC is the feature mapping effect obtaining linear richness and nonlinear richness. Meanwhile, the delay loop increases the MC to improve the forecasting performance. Our proposed algorithm further enriches the nonlinearity of the data in the input layer by adding simple nonlinear node, inspired by the nonlinear vector autoregression (NVAR) machine. Additionally, in order to make our network more suitable for temperature compensation tasks, we increased the MC in the output layer by time multiplexing, presenting a new structure called input–output-improved reservoir computing (IOI-RC).

As shown in Figure 1b, the input u(t_i_) first goes through a polynomial nonlinear node (quadratic in this paper) before being broadcast by the **mask**. We put the nonlinear transform before the mask to ensure masking consistency. In other words, if we square the **x_lin,i_** after masking, the **mask** is multiplied twice in the nonlinear vector **x_nl,i_**, which is not the interested data and reduces system performance. The total input feature vector **x_i_** is 2N-dimensional, and the time interval between each element stays the same θ, matching with the resonator decay time (T_d_). The reservoir layer of IOI-RC is similar to traditional RC. After two nonlinear transformations, the system obtains reservoir feature vector **r_i_** with huge linear and nonlinear richness so that the regression process catches more dynamic information. As the **r_i_** consists of single-nonlinear elements **r_in_** and double-nonlinear elements **r_iqn_**, we found that adding the two different kinds of elements through the delay loop would enhance the system’s performance, which is calculated by:(3)$$\left\{\begin{array}{c} r_{in\;}=\mathrm{DF}\left({\alpha r}_{\left(i-1\right)qn\;}{+\;\beta x}_{in}\right),\;1\;\leq \;n\;\leq \;N.\\ r_{iqn}=\mathrm{DF}\left({\alpha r}_{in\;}{+\;\beta x}_{iqn}\right),\;1\;\leq \;n\;\leq \;N. \end{array}\right.$$
where DF represents the duffing transform, subscript i represents the t_i_ moment, subscript n represents the element position, α is the feedback gain, and β is the input gain. In this way, the delay time remains: τ = Nθ. In order to achieve real-time compensation, the data sampling interval should be small, much shorter than the hysteresis time. As a result, the system can only receive short-term MC. As illustrated in Figure 1b, present mapping feature vector **r_i_** is concatenated by several previous vectors **r_i-sk_**, where s is a positive integer and k is the time step. The product sk is matched with the correlation length of adjacent data, which represents the needed MC. While the choose of s is quite flexible, which selects how many past data points are needed, time step k is determined by how long ago the data most relevant to the present occurs (lagging time of hysteresis for dynamic temperature compensation), which is related to a specific task. Initially, the total output feature **o_i_** has a length of 2(s + 1)N as well as the weight vector **w**, but we can retain a part of element of **o_i_**, which balances processing speed and accuracy. The training method remains a ridge regression, with the same target value of a traditional RC.

IOI-RC is theoretically more time-consuming with an extra nonlinear transform and MC reinforce, but brings much more accuracy. It combines features of NVAR and RC and has the same number of parameters as traditional RC to be optimized. For temperature compensation, as the input temperature data directly flows into the resonator during the experiment and the RC takes almost no time (far less than sampling time), we can obtain real-time compensation.

### 2.2. Temperature Compensation Method

The schematic of our MEMS resonant accelerometer is shown in Figure 2. The length of our encapsulated device is 6950 μm, the width is 5300 μm and the thickness is 450 μm.

It is fabricated by standard silicon processing. There is a double-ended tune fork resonator with one end connected to a proof mass by a pair of micro-lever force amplifier. It has a scale factor of 860 Hz/g. Our device is capacitive driven and sensed by electrodes at two sides of the resonator. The proof mass has four suspended cantilevers around it acting as structural supports. When an acceleration is applied to the proof mass, inertial force is generated and amplified by the micro-lever. The resonator is stretched or compressed by the inertial force, causing stiffness change and resulting in frequency drift (FD).

When the ambient temperature changes rapidly, the thermal gradient generated by the uneven temperature distribution on the accelerometer causes the hysteresis of its temperature response, which brings the classic f-T modelling methods unacceptable measurement error. This dynamic error that contains a temporal and logical relationship could not be expressed as a fitting function directly. Therefore, we innovatively introduced RC, which is best-in-class for predicting chaotic systems, combined with classic polynomial fitting to deal with this complicated prediction task. Considering that the FD caused by the change in scale factor of our home-made accelerometer is negligible under changing temperature, only the zero-bias FD for the whole acceleration range is supposed to be taken into account in our compensation model, which is calculated by:(4)$$\frac{\Delta f}{f_{0}}=\mathrm{cT}+d(T).$$
where cT represents the static polynomial fitting, and d(T) is the residual dynamic drift that would be predicted by RC. We chose T rather than ΔT because of the desired positive normalized input data for the RC system, as the effective information for amplitude modulation is the absolute value.

Figure 3 illustrates our compensation method for the corresponding f-T model. The system respectively yields static FD (FD_s_), and dynamic FD (FD_d_). They are added up as the total estimated FD. This two-step method better accords with our intuition, as it first finds the static operation point of the resonator in a given temperature, and then predicts the dynamic characteristics of the whole system. Furthermore, the RC based on MEMS is hardware implemented with a similar fabrication to the accelerometer, which may be possible to integrate, affording a sensor system with online compensation.

### 2.3. Experiment

Figure 4a shows the schematic of our temperature experiment’s setup. Briefly, the MEMS accelerometer was fixed horizontally with zero acceleration in the probe station (Lakeshore Model TTPX, Lake Shore Cryotronics, Inc., Westerville, OH, USA). Heating was controlled using a temperature controller (Lakeshore Model 336, Lake Shore Cryotronics, Inc.) with a heater and a temperature sensor placed near the accelerometer. Cooling was carried out by liquid nitrogen and the flow rate is fixed so that the equivalent cooling power was about half of the max heating power. The temperature controller can set the temperature value and implement stabilization using a PID algorithm. It can also set temperature ramping rate up to 50 °C/min, achieving various temperature changes. The accelerometer interface circuit was connected outside the probe station in order to eliminate the influence of temperature on the circuit. It is a closed-loop circuit for real-time driving and sensing and is powered by a DC power source (KEITHLEY 2450, Keithley Instruments, Cleveland, OH, USA). The resonator frequency is monitored using a frequency counter (KEYSIGHT 53230A, Keysight, Santa Rosa, CA, USA). Temperature and frequency data were collected using a computer. Figure 4b shows a real picture of the experiment.

Our temperature experiment has two steps as shown in Figure 5: static step and dynamic step, which, respectively, yield FD_s_ and FD_d_ for the f-T model in Figure 2. We carried out the experiment on two kinds of model separately. We first calibrated static temperature data for polynomial fitting (5th order) with each point stable for an hour, and then collected data from the dynamic experiment to train the IOI-RC. Our training dataset (TrDs) started from a −10 °C stabilization and contained plenty of up–down processes with multiple trends (shown in the next section). Two testing datasets (TeDs1, TeDs2) were designed, while TeDs1 was collected continuously behind TrDs, and TeDs2 was collected after another −10 °C stabilization in order to validate long-term effectiveness of the system. The range of temperature and the ramping rate were −20~60 °C and −30~25 °C/min, respectively. As explained before, the compensation error is calculated using FD-FD_s_-FD_d_.

## 3. Results

### 3.1. NARMA10 Task

In order to verify our IOI-RC before the temperature experiment, we simulated the well-known NARMA10 task in the RC community [26]. We generated 1000 time points for training and 1000 time points for testing. Three different algorithms are compared for the NARMA10 task: ESN, T-RC, and IOI-RC. The basic structures are similar, and we set the number of virtual nodes as the same for all three algorithms. Except for the improved parts, the IOI-RC parameters were the same as T-RC. We chose N = 50, s = 9, k = 1, and the delay length of 50. Therefore, 10 adjacent points were jointed together in the network to obtain a strong MC specially for this task. We retained 20% of the output vector elements evenly, so a single output vector **oi** contains 200 elements, as well as the **w**.

Table 1 shows the system’s performance. The normalized root mean square error (NMSE) is calculated for evaluation, with IOI-RC being the smallest, which validates the superiority of our optimized algorithm.

### 3.2. Dynamic Temperature Compensation Task

In total, five algorithms are compared: polynomial fitting (PF), PF–ESN, PF–T-RC, IOI-RC, and PF–IOI-RC. We also chose a one-step method with only IOI-RC predicting the whole FD to compare. While PF continued using the 5th order model, the other four algorithms were all developed using TrDs. The ESN was optimized by scaling reservoir states for the active function [13], and the two kinds of RC were hardware-implemented with the same parameters. For IOI-RC, we chose θ = 1.05 ms, N = 50, α = 0.9, β = 1, s = 3, and k = 400. Specifically, θ was set as 0.1 T_d_, s and k were set according to the hysteresis time around 1 min, which covers 1200 samples. As hysteresis is a homogeneous process for all time points, we expanded the output feature vectors by evenly selecting three additional feature vectors among related samples, that is: **o_i_** = **r**_i_ ⊕ **r_i-400_** ⊕ **r_i-80_**_0_ ⊕ **r_i-1200_**. We did not discard elements such as the NARMA10 task for better performance. In this way, the prediction of each time point was trained by data from a past period, so the model could remember dynamic information of the past directly.

The compensation results are shown in Figure 6. Figure 6a shows the prediction result of TrDs and TeDs1. The blue line is the real FD measured before compensation, while the green line and red line stand for the prediction of PF and PF—IOI-RC chosen as examples. Because PF could not catch dynamic information, a given temperature at a different time would yield the same FD_s_, which means the hysteresis could not be compensated. For TeDs1, the residual error after compensation of the five methods is presented in Figure 6b. It is obvious that our proposed method has the best performance, as the residual curve is almost zero. For TeDs2, Figure 6c compares the real and predicted FD curves of all five methods, and Figure 6d shows the residual error. Because TeDs2 is a new run of temperature stabilization, the thermal flow is initialized at the first few time points, which are often called warm-up points. It can be seen that ESN has an obvious dithering area at the beginning with some large error points even beyond the axis. A possible explanation for this unsteadiness is that the reservoir parameters of ESN are randomly selected, so they need more warm-up time. However, our RC structure is more stable for long-term testing, benefitting from its simple network, short warm-up period and large MC.

Table 2 illustrates that the PF–IOI-RC has the best performance among the five methods, with up to 90% improvement in root mean square error (RMSE) and 93% improvement in maximum absolute error (MAE) over uncompensated data. The one-step method can also depict the dynamic f-T relationship but is weaker than the two-step method. This is probably because the PF first finds a near-static operation frequency, then the remaining FD_d_ is predicted by IOI-RC more easily than the origin FD. Although the one-step method has a little bit smaller MAE in TeDs1, it may be due to some individually measured noise points causing PF–IOI-RC a large error. But on the whole, the two-step method is more accurate in terms of RMSE. Compared to other state-of-art works in Table 3, our work shows superiority, which infers that our IOI-RC is well-suited for dynamic temperature compensation. Taken together, these results suggest that IOI-RC has a superb performance in dynamic temperature compensation, which solves the hysteresis problem significantly.

Notably, in the above experiment, we keep the physical reservoir based on the MEMS clamp–clamp beam resonator (details can be referred to in our previous work [22]) at a stationary temperature to ensure its stationary operation point. However, it will be a challenge to achieve online temperature compensation considering that we are supposed to integrate the physical reservoir into the sensors that need temperature compensation in the following work. In our future work, we propose two improvement approaches. One is using a physical system with nonlinear dynamics, which is not sensitive to temperature, such as the well-known Mackey–Glass circuit [27]. Another method is to divide the target temperature region into different intervals, so that the training can be performed segmentally in these intervals based on our IOI-RC. In a small temperature range (for example, 5 °C~10 °C), the RC characteristic could be treated as invariant, so we can separately train several weight vectors **w** for different temperature ranges. Meanwhile, every single **w** will carry some necessary information of the signal fluctuation result from small temperature change by training in the dynamic case. Combining system control via FPGA, the segmented model can primarily provide online compensation under changing temperatures.
micromachines-14-00161-t003_Table 3Table 3Comparation between this work and other reported.ReferenceMethodTemperature Range (°C)Error (%)Improvement (%)[12]PF−20~600.4380[28]AGA-BP−20~600.3885[29]RBF-NN−25~650.6882This workPF–IOI-RC−20~600.01693

## 4. Conclusions

In conclusion, we proposed a temperature compensation method based on the physical RC with a MEMS resonator. With this method, we first achieved temperature compensation for the MEMS accelerometer, which we demonstrated previously [30], under rapid ambient temperature changes and resolved the dramatic hysteresis phenomenon. Multiple analyses revealed that our proposed model of RC with improved structure can achieve high prediction accuracy, decreasing the zero bias up to tenfold with only 114 ppm in a temperature range from −20 °C to 60 °C. The findings of this research provide insights for a novel real-time online temperature compensation method using hardware RC integrated with an accelerometer, which add to the rapidly expanding field of hardware implementation for neural networks, as the RC shows superiority with high accuracy and simple structure. Being limited to the unknown performance of temperature influence of the RC system, we divided data acquisition and processing. Additionally, we are working on exploring the deep mechanism behind the thermal lagging affect’s compensation. Although we cannot provide enough explanation, we have demonstrated that physical RC can be applied for dynamic temperature compensation of MEMS sensors. In our future work, we will focus on simulation work to determine hysteresis quantitively and devote attention to exploring a temperature-independent physical RC system to achieve real-time online temperature compensation for MEMS sensors.

## Figures and Tables

**Figure 1 micromachines-14-00161-f001:**
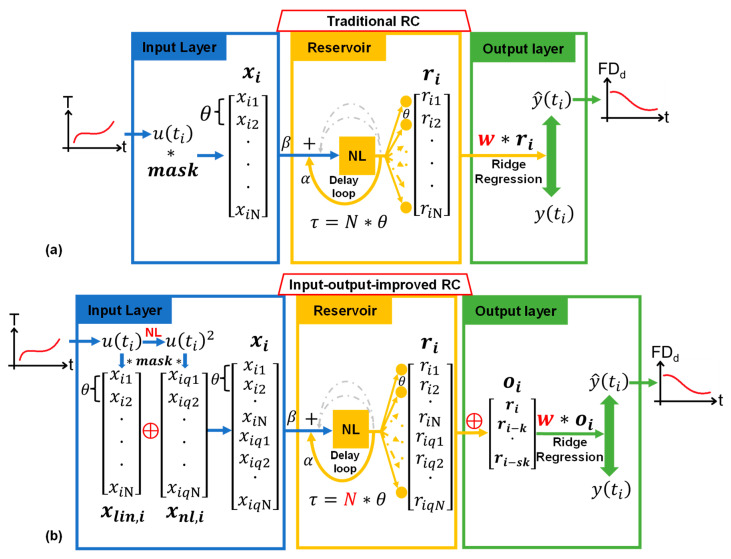
The T-RC and IOI-RC. (**a**) The T-RC structure: the mask signal broadcasts the original data, and the delay loop adds correlation of adjacent time data. (**b**) The IOI-RC structure: input layer increases nonlinear richness and output layer enhances long-term MC.

**Figure 2 micromachines-14-00161-f002:**
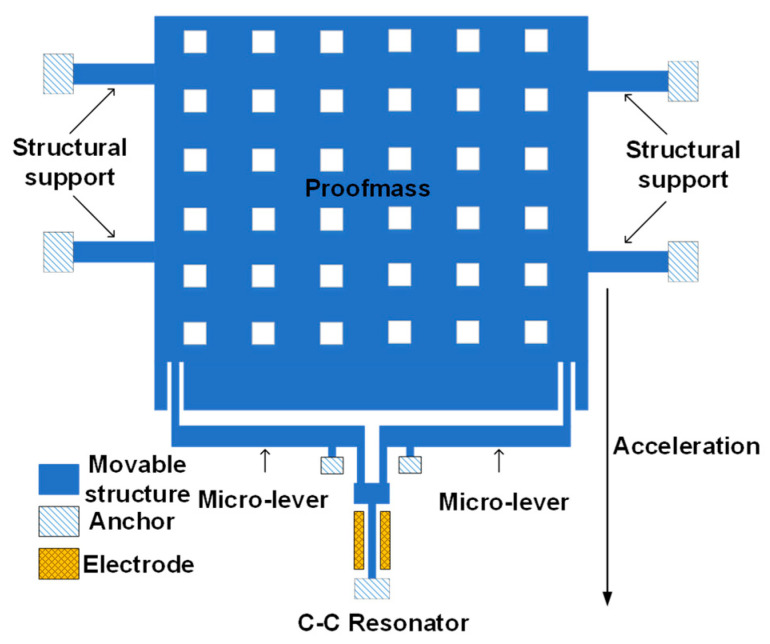
The schematic of our silicon-based MEMS accelerometer. The tune fork resonator is an accelerometer sensor, which is driven and sensed by capacitors. The proof mass has four structural supports and a pair of micro-levers.

**Figure 3 micromachines-14-00161-f003:**
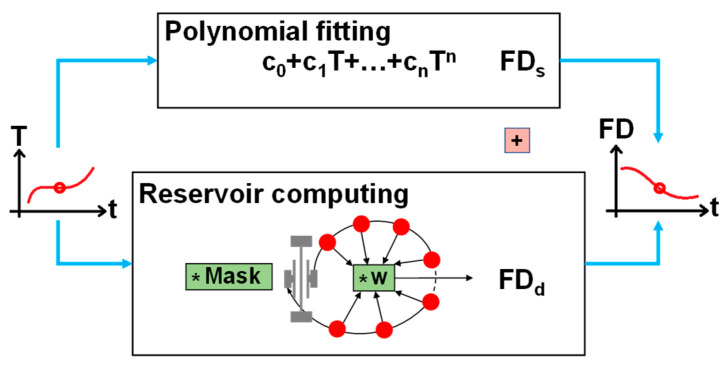
The flow chart of the temperature compensation method. Measured temperature T is sent to polynomial and reservoir simultaneously. The polynomial fitting yields FD_s_ and the reservoir computing predicts the FD_d_. The total estimated FD is the sum of FD_s_ and FD_d_.

**Figure 4 micromachines-14-00161-f004:**
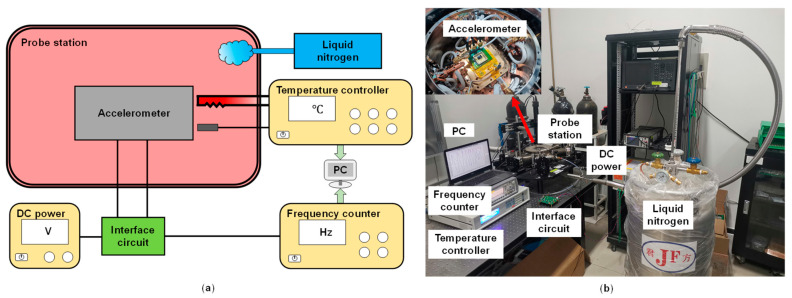
Temperature experiment setup. (**a**) The schematic of the temperature experiment: The probe station act as a temperature oven heated up by the temperature controller and cooled down by liquid nitrogen. The accelerometer was zero-bias and driven by an interface circuit. The frequency counter monitored the frequency changes. (**b**) The real picture of the temperature experiment.

**Figure 5 micromachines-14-00161-f005:**
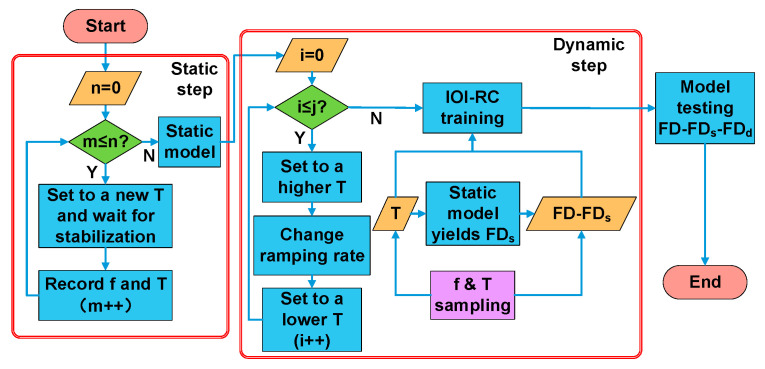
Flow chart of the temperature experiment steps. Max–min temperature values as well as the ramping rate are quite varied, which makes the dataset diverse. Sampling temperature goes through the static model and the output is sent to IOI-RC, and then the dynamic model is trained and tested.

**Figure 6 micromachines-14-00161-f006:**
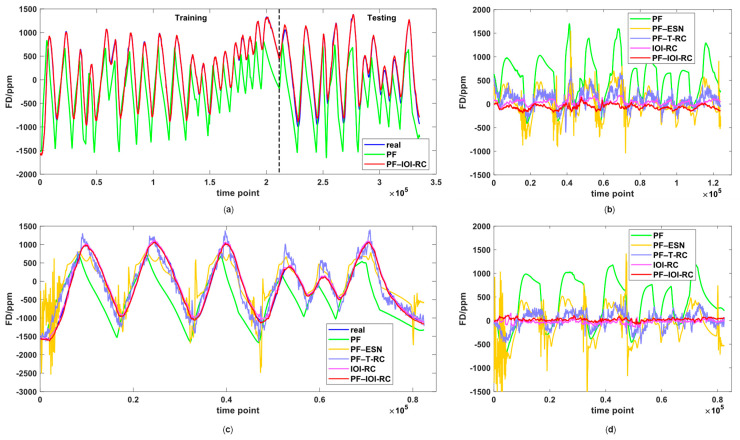
Frequency prediction curves and residual error curves of the three datasets. (**a**) The real and predicted FD of TrDs and TeDs1. (**b**) The remaining FD error after compensation of TeDs1. (**c**) The real and predicted FD of TeDs2. (**d**) The remaining FD error after compensation of TeDs2.

**Table 1 micromachines-14-00161-t001:** Performance comparation of NARMA10 task.

Methods	NMSE
ESN	0.3045
T-RC	0.1142
IOI-RC	0.0852

**Table 2 micromachines-14-00161-t002:** Performance comparation of dynamic temperature compensation task.

Method	TeDs1		TeDs2	
	RMSE (ppm)	MAE (ppm)	RMSE (ppm)	MAE (ppm)
raw	651	1580	715	1580
PF	701	1435	627	1233
PF–ESN	328	1550	423	1974
PF–T-RC	290	769	249	707
IOI-RC	165	146	96	165
PF–IOI-RC	132	164	76	114

## Data Availability

Not applicable.

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
