# Peer review of "Input–Output-Improved Reservoir Computing Based on Duffing Resonator Processing Dynamic Temperature Compensation for MEMS Resonant Accelerometer"

_micromachines, 2023, doi:10.3390/mi14010161_

Round 1
Reviewer 1 Report
This statement needs more explanation, maybe an example could be helpful: “Our method is universal and easy for hardware implementation.”
The explanation of their algorithm is not clear when connected to figure 1 b. Moreover, how they arrive at equation (1) is not clear.
This statement is not clear: “We chose T rather than ΔT because of the desired positive 169 normalized input data for the RC system.”
Needs more explanation about the DAQ used in your experiments such as the sampling rate and other stuff, it is not enough just to say by computer.
They are many repeated statements such as “Three different algorithms are compared for 245 NARMA10 tasks: ESN, T-RC and IOI-RC.” Repeated twice in the NARMA10 task section.
Too many confusing terminologies in the paper: PF&ESN, PF&T-RC, 260 IOI-RC, and PF&IOI-RC
The dynamic and static FD needs more explanation.
The whole discussion after Figure 5 and before the Results section is not clear and need major revision
in figure 1b there was two input u(t) and u(t)2 , the author didn’t mention clearly u(t)2 , why and how he used it in the algorithm.
The result part should be improved, because there are a lot of different algorithms involved the reader misses the relation between different parts
There are some repeated sentences in the conclusion.
Reviewer 2 Report
This paper proposes a reservoir computing (RC) structure based on a nonlinear silicon resonator. The RC is used to compensate the dramatic hysteresis phenomenon of temperature response for MEMS accelerometer under rapid ambient temperature changing, achieving high accuracy. It is interesting to those who are working in this field. However, the writing and organization of this paper need improving. For example, in “4. Conclusions”, the sentence “These results add to the rapidly expanding field of hardware implementation for neural network” is repeated two times, etc.
Reviewer 3 Report
The dependence of MEMS resonant accelerometer on temperature is very sensitive. Facing the difficulties of existing classic temperature compensation methods, authors propose a novel reservoir computing structure based on a nonlinear silicon resonator which is specifically improved for predicting dynamic information that is referred to as the input-output-improved reservoir computing (IOI-RC) algorithm. In addition, in their scheme, they combine the polynomial fitting with the reservoir computing on the input data mapping ensuring, and this combination make the system always reside in the rich nonlinear state, which overcomes the disadvantages of existing classic temperature compensation methods. This manuscript also gives some good experimental results. Authors report a scheme of feasibility of realizing dynamic temperature compensation precisely, which provides a potential real-time online temperature compensation method and a sensor system with edge computing. Their results are interesting and convincing, so I would like to recommend it for publication in Micromachines.
